# Categorisation of Patients’ Anticholinergic Burden at Admission and Discharge from the Geriatric Ward of Sønderjylland Hospital

**DOI:** 10.3390/pharmacy12060160

**Published:** 2024-10-25

**Authors:** Cecilie Marie Bæk Kehman, Maja Schlünsen, Lene Juel Kjeldsen

**Affiliations:** 1Clinical Pharmacy, Capital Region’s Pharmacy, 3400 Hillerød, Denmark; cillekehman@hotmail.com; 2Institute for Regional Health Research, The University of Southern Denmark, 5230 Odense, Denmark; lene.juel.kjeldsen@rsyd.dk; 3Hospital Pharmacy Research Unit, University Hospital of Southern Denmark (Hospital Sønderjylland), 6200 Aabenraa, Denmark

**Keywords:** anticholinergic burden scale, hospital admission, geriatric patients, cross-sectional study

## Abstract

Background: High anticholinergic burden is associated with an increased risk of hospitalisation, readmission, and mortality in geriatric patients. The objectives were to develop an updated anticholinergic burden scale for drugs registered in Denmark and to estimate the burden at admission and discharge for hospitalised patients at the Geriatric Ward of Sønderjylland Hospital. Methods: The updated scale was developed through a systematic evaluation of the anticholinergic effect for all active pharmaceutical ingredients (APIs) listed on validated burden scales. APIs registered in 2020 and 2021 were evaluated separately for possible anticholinergic effect. The anticholinergic effect of each API was scored from 1 (low) to 3 (high). The scale was applied to medical records for patients hospitalised between October 2021 and March 2022. Results: The scale comprised 87 APIs with anticholinergic effect. We applied the scale on 196 patients aged (median [IQR]) 84 (78–89) years. Of these patients, 75 (38.3%) had a high burden (≥3) on admission. These patients had significantly higher drug use and higher risk of 30-day readmission but no relationship with length of stay. Overall, the anticholinergic burden was unchanged at discharge for 109 (55.1%) patients. Conclusion: An updated scale for estimation of the anticholinergic burden in geriatric patients was successfully developed, and a high burden among the admitted geriatric patients was found.

## 1. Introduction

A high anticholinergic burden is associated with an increased risk of admission, readmission, and mortality in geriatric patients [1,2,3]. Due to frailty, comorbidities, polypharmacy, and physiological changes, geriatric patients are at risk of experiencing anticholinergic side effects such as confusion, orthostatic hypotension, tendency to fall, and obstipation [4]. 

The total anticholinergic activity of a patient’s drug treatment constitutes the patient’s anticholinergic burden. An anticholinergic burden scale may help identify and assist in preventing the use of inappropriate anticholinergic drugs through burden estimation. The burden is a cumulated score expressing the patient´s risk of developing anticholinergic side effects. When applying the Anticholinergic Cognitive Burden scale, Sørensen et al. (2022) found that 19.3% of Danish geriatric patients had a high anticholinergic burden on admission to hospital [1]. Other studies showed that the burden increased for one-fifth of patients during hospitalisation [2,5]. The scales also help identify suboptimal drug use, especially among patients with dementia [5,6]. Hence, the risks associated with anticholinergic burden remain a challenge among geriatric patients; however, no updated scale exists to assist in identifying and measuring the size of the burden.

The objectives of this study were to develop an updated anticholinergic burden scale for drugs registered in Denmark, and to estimate the burden at admission and discharge for patients admitted to the Geriatric Ward at Sønderjylland Hospital.

## 2. Materials and Methods

### 2.1. Development of the Updated Scale

We conducted a systematic literature search in PubMed and Embase to identify existing anticholinergic burden scales. Anticholinergic burden scales, which were validated, were selected for the study. They included the following: Anticholinergic Risk Scale, Anticholinergic Drug Scale, Anticholinergic Cognitive Burden, Anticholinergic Activity Scale, German Anticholinergic Burden Score scale, CRIDECO Anticholinergic Load Scale, and Anticholinergic Loading Scale [7,8,9,10,11,12,13,14]. The current Danish scale from the Institute for Rational Pharmacotherapy from 2017 was also included [4]. All included scales categorised an estimated burden at ≥3 as clinically relevant.

Active pharmaceutical ingredients (APIs) listed on the scales and registered in Denmark were systematically evaluated for anticholinergic activity.

APIs appearing both on one or more of the following scales: Rational Pharmacotherapy scale, CRIDECO Anticholinergic Load Scale, and German Anticholinergic Burden Score scale; and on two or more other scales were included on the new scale. APIs appearing either on only one of the following scales: Rational Pharmacotherapy scale, CRIDECO Anticholinergic Load Scale, or German Anticholinergic Burden Score scale; or on two or more other scales were also evaluated for clinical and biological relevance.

Clinical relevance was defined as four or more different anticholinergic side effects according to the summaries of product characteristics. The biological relevance was defined as relevant binding affinities to muscarinic receptors listed in MICROMEDEX, PubChem, or original articles. Formulations with non-systemic absorption were excluded.

The APIs were categorised according to low (score 1), moderate (score 2), or high (score 3) anticholinergic effect and scored based on the score from the included scales. If there were discrepancies, then the score was given as the mean´s nearest integer. APIs registered in 2020 and 2021 were evaluated for clinical and biological relevance according to the above-mentioned definitions.

### 2.2. Clinical Assessment

This study was conducted as a cross-sectional study at a Geriatric Ward with 22 beds in Sønderjylland Hospital in Denmark. The study included patients admitted from 1 October 2021 to 30 March 2022. Terminal patients and patients passing away during hospitalisation were excluded. The primary outcome was the patients’ anticholinergic burden at admission and discharge, and secondary outcomes included length of stay and 30-day hospital readmission.

#### 2.2.1. Data Collection

Data were collected from patients’ electronic medical records at the index admission during the study period. The collected data included age, gender, cause of hospitalisation, length of stay, comorbidities at discharge, and 30-day hospital readmission. All medicine lists were collected from registered medication reconciliations and/or prescriptions at admission and previous hospitalisations. Medicine used pro re nata were included in the burden estimation. Prescribed vitamins and minerals used for deficiency or prophylaxis were included as “general drug use”.

#### 2.2.2. Data Analysis

Patients were categorised into four groups: no (score 0), low (score 1), moderate (score 2), and high (score ≥ 3) burden. Categorical variables were expressed as numbers and percentages, the numeric variables as median and interquartile range (IQR). The groups were compared using Fisher’s exact test for categorical variables and the Mann–Whitney U test for numeric variables. McNemar’s test was used to assess anticholinergic drug use and anticholinergic burden between admission and discharge. *p* values < 0.05 indicated statistical significance.

### 2.3. Ethics

This study was not a biomedical study, and thus approval from the Danish National Ethic Committee was not required. The study was approved by the hospital management. To ensure data confidentiality, the data were anonymised and collected on site.

## 3. Results

### 3.1. Updated Scale

Based on the conducted systematic literature search, 87 APIs out of 198 evaluated were included in the updated scale (Table 1). For 28 APIs, the score was recalculated. In total, 38 (43.7%) APIs were identified as low-activity, 23 (26.4%) APIs as moderate-activity, and lastly, 26 (29.9%) APIs as high-activity. Forty-eight APIs (55.1%) were APIs affecting the central nervous system (therapeutic group N, Table 1).

### 3.2. Clinical Assessment

#### 3.2.1. Baseline Characteristics

Electronic medical records for 196 patients were reviewed. Of the included patients, 75 (38.3%) had a high anticholinergic burden at admission (Table 2). The prevalence of chronic obstructive pulmonary disease (COPD), both as comorbidity and as a cause of hospitalisation, was significantly higher for patients with a high anticholinergic burden compared to patients with no burden (Table 2). In addition, 17 patients (8.7%) were admitted due to fall tendency or fall-related fracture, but we found no statistically significant differences in admission due to falls or fall-related fractures between the groups. Patients with an anticholinergic burden had significantly more prescriptions of medicine.

The proportion of patients prescribed salbutamol, pantoprazole, sodium picosulfate, and osmotic laxatives was significantly higher for patients with a high burden compared to those without. Among the 19 patients who were treated with cholinesterase inhibitors, 14 patients (73.7%) had an anticholinergic burden.

#### 3.2.2. Anticholinergic Drug Use

Of the 87 APIs on the scale, 43 (48.9%) were represented in the study population at admission, with low-activity anticholinergic APIs contributing the most (n = 22; 51.2%). Furosemide (35.2%), oxycodone (12.8%), and sertraline (11.2%) were among the most prescribed. The proportion of patients prescribed mirtazapine and morphine was significantly higher for patients with high anticholinergic burden compared to patients without (*p* = 0.005; 0.002). Tramadol was used by 18.7% and 5.95% patients with high and moderate anticholinergic burden, respectively (*p* = 0.052). At discharge, tramadol prescriptions were significantly reduced by discontinuation (n = 5), shift to morphine (n = 3), or morphine dosage adjustment (n = 1).

#### 3.2.3. Anticholinergic Burden at Discharge

Upon discharge, the number of patients experiencing no anticholinergic burden decreased to 39 (19.9%), while that of patients with a high anticholinergic burden increased to 79 (40.3%). These changes were not statistically significant compared to the respective proportions at admission (*p* = 0.108 and *p* = 0.627). The changes in burden from admission to discharge, stratified by the different anticholinergic burden found in the study population, are presented in Figure 1. About half of the patients (n = 109; 55.1%) had an unchanged anticholinergic burden at discharge (Figure 1)**.** Among the 52 (26.5%) patients with an increased burden at discharge, the burden of 21 (40.3%) patients became clinically relevant. Of the 31 patients with a very high burden (≥5), 15 patients had a reduced burden at discharge (48.4%).

When the pharmacy technicians conducted the prescription reviews, they also identified potential suboptimal medication treatments, which were solved in collaboration with the ward staff with the aim of increasing the quality of the individual medication treatment.

#### 3.2.4. Length of Stay and 30-Day Hospital Readmission

Length of stay was not influenced by the anticholinergic burden (Table 3). The prevalence of 30-day hospital readmission was significantly higher for patients with low and high burden compared to patients with no burden (Table 3).

## 4. Discussion

The updated anticholinergic burden scale contained 87 APIs. When applied to the 196 medical charts of patients admitted to the Geriatric Ward, we found that 75 (38.3%) had a high anticholinergic burden (≥3) on admission. These patients had significantly higher drug use and a higher risk of readmission within 30 days. The burden was reduced during hospital admission primarily for patients with a high anticholinergic burden, but it was unchanged at discharge for 109 (55.1%) patients.

Sørensen et al. (2022) found that 19.3% of geriatric patients had a high burden, and in other European studies, the percentage was between 14% and 26.8% [1,15,16]. The differences between this study and the literature can be explained by the higher median age, smaller study population size, and the application of an adjusted scale in this study. Like our result, Sørensen et al. (2022) found that APIs with low anticholinergic activity were most prescribed (88.1%) [1]. The high proportion of anticholinergic burden in these studies can perhaps be attributed to the lack of knowledge in the clinic regarding anticholinergic activity. Another explanation could be the lack of evidence in the literature about API anticholinergic activity, and thus a lack of consensus about anticholinergic activity may occur.

Laxatives, in particular, were prescribed for patients with a high anticholinergic burden. Many APIs with an anticholinergic activity can cause constipation as a side effect, and hence the prescription of laxatives could indicate a “prescribing cascade” [17]. The same results were found in an Italian study at a nursing home [18]. Here, the patients who were prescribed antidepressants, anti-Parkinson dopaminergic agents, and benzodiazepines were also prescribed laxatives [18]. Optimising the prescription of APIs with anticholinergic burden might help reduce the potential “prescribing cascades” with laxatives.

Tramadol is not recommended for geriatric patients due to limited effect, risk of side effects, and dependency, just like other opioids [19,20]. Although a high proportion of patients were treated with tramadol at admission, at discharge, this was the only API where the number of prescriptions was significantly reduced. This could indicate a focus on inappropriate medication during hospitalisation. Furthermore, this is in accordance with international and national clinical guidelines [21,22,23].

Like our result, studies on American and Italian geriatric patients found that the anticholinergic burden was unchanged for, respectively, 63.8% and 49.1% of patients at discharge [2,24]. This might be due to lack of knowledge, down-prioritisation, or unindicated medicine changes. The burden was most often reduced for patients with a very high burden. It is unknown if knowledge of the anticholinergic burden caused the decrease. Changes in patients’ therapies could be due to medication reviews with a focus on deprescribing ineffective or irrelevant medicine such as urologicals or antiemetics [23]. Another limitation is any failed deprescribing or planned tapering as it is not expressed in the burden at discharge.

We found that more patients with a burden (low or high) were readmitted after 30 days than patients without a burden. Rice et al. (2021) found an association between a high anticholinergic burden and 30-day hospital readmission (*p* < 0.001) [2]. However, large studies adjusting for age, sex, and comorbidities show no association [25].

From a clinical perspective, this study can help increase the focus on the anticholinergic burden among geriatric patients. To our knowledge, this is the first Danish study suggesting a new anticholinergic scale and applying it. This scale can be used to help highlight possible side effects and indications for deprescribing or dosage adjustments. The prescribing physician could use the scale as guide to prescribe APIs with low or no burden, especially for polypharmacy patients. This would, for example, be relevant when prescribing antihistamines or antipsychotics. For patients where anticholinergic APIs are needed, the scale can help highlight the need for the monitoring or prophylaxis of side effects [4].

This scale can also help identify the irrational use of medicine. In the present study, we found that three-quarters of the dementia patients were treated with cholinesterase inhibitors. These patients risk treatment failure as anticholinergic APIs block the effect of the cholinesterase inhibitors. The central anticholinergic side effects can also advance cognitive decline in dementia [11,16].

### Limitations

This scale might not be exhaustive for all anticholinergic APIs in Denmark [4,7,14]. Due to our inclusion criteria, relevant APIs such as APIs with limited reported side effects might have been excluded. Another limitation is patient relevance. As expected, a high proportion of COPD patients in this study had a high anticholinergic burden, but an essential part of the treatment regime for COPD is inhaled muscarinic antagonist [26]. For these patients, anticholinergic burden is not an indication for alternative treatment. A similar issue applies to patients with psychiatric disorders.

For many of the patients, compliance was unknown, which meant the inclusion of possible non-used drugs and irrelevant *pro re nata* drugs. This can have resulted in an overestimation of the burden and number of drugs. The comorbidities were only based on the registered diagnosis, but the validity of diagnosis coding may be a subject for uncertainty [27,28,29].

## 5. Conclusions

This study successfully developed a new scale for the estimation of anticholinergic burden in Danish geriatric patients. When applying the scale, we found a high anticholinergic burden among the admitted geriatric patients, and the burden remained unchanged at discharge for more than half of the patients. Further studies are needed to investigate the health-related benefits of reducing the burden. 

## Figures and Tables

**Figure 1 pharmacy-12-00160-f001:**
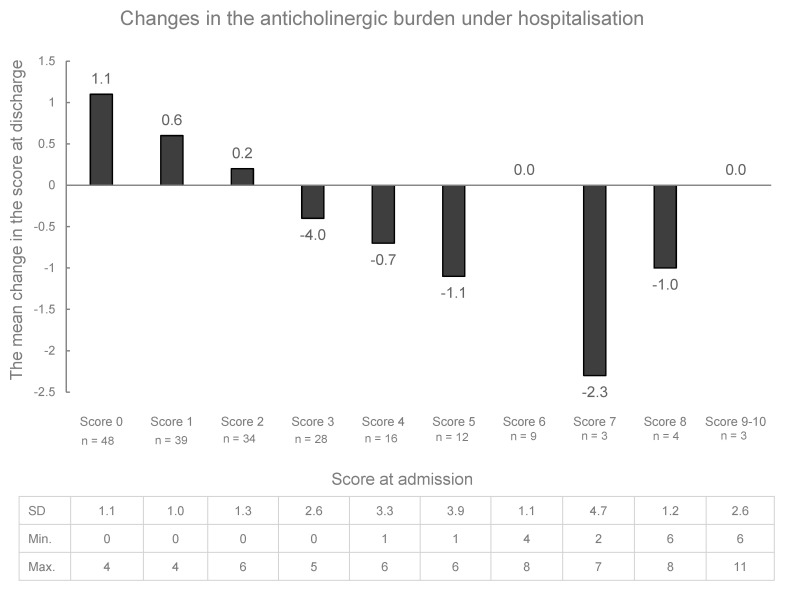
The mean change in the anticholinergic burden between admission and discharge. The standard deviation (SD), minimum, and maximum for each patient group are represented at the bottom of this figure.

**Table 1 pharmacy-12-00160-t001:** The new anticholinergic scale for the estimation of geriatric patients’ anticholinergic burden. N = 87.

Therapeutic Group		Score 1 Low Activity	Score 2 Moderate Activity	Score 3 High Activity
A03	Drugs for functional gastrointestinal disorders	Metoclopramide		Atropine Propantheline
A04	Antiemetics and antinauseants			Hyoscin butylbromide Hyoscin Hyoscyamine
A07	Antidiarrheals, intestinal anti-inflammatory/anti-infective agents		Loperamide	
B01	Antithrombotic agents	Dipyridamole		
C01	Cardiac therapy	Digoxin		
C03	Diuretics	Furosemide		
C07	Beta blocking agents	Chlorthalidone		
C08	Calcium channel blockers	Nifedipine Diltiazem ^1^		
C09	Agents acting on the renin–angiotensin system	Captopril		
G04	Urologicals			Darifenacin Fesoterodine Oxybytynin Solifenacine Trospium Tolterodine
H02	Corticosteroids for systemic use	Hydrocortisone Prednisone		
J01	Antibacterials for systemic use	Gentamicin		
M03	Muscle relaxants		Baclofen	Tizanidine
N02	Analgesics	Codeine Fentanyl Morphine Oxycodone Tapentadol	Tramadol	
N03	Antiepileptics	Clonazepam Valproate	Carbamazepine Oxcarbazepine	
N04	Anti-Parkinson drugs	Bromocriptine Carbidopa/Levodopa Entacapone Pramipexole Rotigotine Selegiline	Amantadine	Biperiden Orphenadrine Procyclidine
N05	Psycholeptics	Alprazolam Asenapine Diazepam Lorazepam Oxazepam Risperidone Ziprasidone	Haloperidole Olanzapine Pimozide Quetiapine Zuclopenthixol ^1^	Clozapine Hydroxyzine Perphenazine
N06	Psychoanaleptics	Citalopram Escitalopram Fluoxetine Fluvoxamine Mirtazapine Reboxetine Sertraline	Dosulepin Paroxetine	Amitriptyline Clomipramine Imipramine Nortriptyline
N07	Other nervous system drugs		Methadone	
R03	Drugs for obstructive airway diseases		Aclidinium Glycopyrronium bromide Ipratropium Theophylline Tiotropium Umeclidinium	
R06	Antihistamines for systemic use		Cetirizine Desloratadine Loratadine	Clemastine Cyclizine Diphenhydramine Meclizine Promethazine

^1^ Including Zuclopenthixol acetate and Zuclopenthixol decanoate.

**Table 2 pharmacy-12-00160-t002:** Baseline patient characteristics stratified on the anticholinergic burden. Significant *p* values are in bold.

	Total	No Burden (Score 0)	Low Burden (Score 1)		Moderate Burden (Score 2)		High Burden (Score ≥ 3)	
				*p* Value		*p* Value		*p* Value
Number of patients, n (%)	196 (100)	48 (24.5)	39 (19.9)	-	34 (17.3)	-	75 (38.3)	-
Female, n (%)	105 (53.6)	24 (50)	16 (41.0)	0.517	18 (52.9)	0.826	47 (62.7)	0.193
Age, years (IQR)	84 (78–89)	84 (80–90)	86 (77.5–88.5)	0.697	85 (81.3–88)	0.906	82 (76–88.5)	0.136
Comorbidity, n (%) Atrial fibrillation/flutter Dementia Hypertension COPD Osteoporosis Type 2 diabetes Malaise	58 (29.6) 28 (14.3) 74 (37.8) 25 (12.8) 36 (18.3) 31 (15.8) 29 (4.6)	9 (18.8) 7 (14.6) 20 (41.7) 2 (4.2) 8 (16.7) 9 (18.8) 9 (18.8)	9 (23.1) 5 (12.8) 14 (35.9) 1 (2.6) 8 (20.5) 8 (20.5) 5 (12.8)	0.791 1.00 0.661 1.00 0.782 1.00 0.563	14 (41.2) 7 (20.5) 14 (41.2) 3 (8.8) 6 (17.6) 2 (5.9) 4 (14.7)	**0.045** 0.557 1.00 0.644 1.00 0.111 0.543	26 (34.7) 9 (12) 26 (34.7) 19 (25.3) 14 (18.7) 12 (16) 11 (14.7)	0.067 0.785 0.451 **0.003** 0.815 0.807 0.620
Total number of drugs (IQR)	11 (7–13)	6.5 (3.3–10)	10 (7–11)	<**0.001**	9 (7–12.8)	<**0.001**	13 (11–17)	<**0.001**
Cause of hospitalisation, n (%) Delirium COPD Pneumonia UTI Other infection	12 (6.1) 8 (4.1) 41 (20.9) 17 (87) 34 (17.3)	3 (6.3) 0 (0) 10 (20.8) 5 (10.4) 5 (10.4)	4 (10.3) 0 (0) 8 (20.5) 2 (5.1) 9 (23.1)	0.700 - 1.00 0.452 0.087	1 (2.9) 0 (0) 8 (23.5) 3 (8.8) 10 (29.4)	0.638 - 0.792 1.00 **0.042**	4 (5.3) 8 (10.7) 15 (20) 7 (9.3) 10 (13.3)	1.00 **0.022** 1.00 1.00 0.780

Interquartile Range (IQR); Chronic Obstructive Pulmonary Disease (COPD); Urinary Tract Disease (UTI).

**Table 3 pharmacy-12-00160-t003:** Length of stay and 30-day readmission for the different patient groups.

	No Burden (Score 0)	Low Burden (Score 1)		Moderate Burden (Score 2)		High Burden(Score ≥ 3)	
			*p* Value		*p* Value		*p* Value
Length of stay, days (IQR)	7 (5–9)	7 (6–9)	0.114	6.5 (5–11)	0.660	7 (5–9)	0.841
30-day readmission, n (%)	8 (16.7)	14 (35.9)	0.049	12 (35.3)	0.069	28 (37.8)	0.015

## Data Availability

The data supporting the results are included in the article, and hence no further data are available.

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
