# Peer review of "Categorisation of Patients’ Anticholinergic Burden at Admission and Discharge from the Geriatric Ward of Sønderjylland Hospital"

_pharmacy, 2024, doi:10.3390/pharmacy12060160_

Round 1
Reviewer 1 Report
Comments and Suggestions for Authors
The manuscript "Categorisation of patients’ anticholinergic burden at admission 2 and discharge from the Geriatrics ward at Hospital Sønderjyl-3 land" is interesting, clear and well written. I have only two issues I would like the authors to address:
1. Figure 1 is very difficult to follow. I tried to add up the number of participants described in the text referring to Figure 1 and the numbers in the Figure and I did not succeed. Could you please look it up again and make it clearer to follow.
2. You assessed the 30 day readmission to hospital. However, readmissions could have been caused by other factors rather than anticholinergic burden. Did you evaluate the reasons for readmission and the causal relationship with anticholinergic burden?
Author Response
The manuscript "Categorisation of patients’ anticholinergic burden at admission and discharge from the Geriatrics ward at Hospital Sønderjylland" is interesting, clear and well written. I have only two issues I would like the authors to address:
Comment 1: Figure 1 is very difficult to follow. I tried to add up the number of participants described in the text referring to Figure 1 and the numbers in the Figure and I did not succeed. Could you please look it up again and make it clearer to follow.
Response 1: Thank you for your comment. We have updated the text supporting Figure 1 to better explain the context of the data presented, line 157-162. Please note that the numbers in Figure 1 and the data presented in the text support each other.
Comment 2: You assessed the 30 day readmission to hospital. However, readmissions could have been caused by other factors rather than anticholinergic burden. Did you evaluate the reasons for readmission and the causal relationship with anticholinergic burden?
Response 2: The reasons for readmissions were not quantified due to the relative low number of hospital readmissions.
Reviewer 2 Report
Comments and Suggestions for Authors
Brief Summary: This is a process improvement project in which an updated scale was developed to estimate the anticholinergic burden of hospitalized geriatric patients in Denmark. Secondary aims were to examine if any change occurred in this burden at discharge, and the relationship to length of stay and 30-day hospital readmission.
General Comments: This manuscript flowed logically and is generally well written with one major concern and a few minor comments. The major concern is the lack of comment about the length of stay and anticholinergic burden which is listed a secondary outcome measure.
Specific Comments
1. Abstract is clear but should also include the length of stay finding.
2. Introduction is clear and logical concluding with the objectives of the study: burden at admission and discharge (and any change), length of stay, and 30-day readmission.
3. Materials and Methods:
a. Development of the Updated Scale: This process is adequately described and appropriate literature used to create the list of 87 drugs. This reviewer was particularly impressed with the inclusion of receptor binding affinities criterion and other descriptions of how the 87 drugs were selected. This alone makes the scale clinically relevant to the institution where used. The simplicity of the scale also is appealing as a clinically relevant tool.
b. The Clinical Assessment: Appropriate and clear participant selection and exclusion as well as the primary and secondary outcomes to be measured.
c. Data Analysis: Appropriate for level of data.
4. Results section with tables
a. The updated scale: Table 1 is clear and concise as are the comments about it.
b. The Clinical Assessment results are clustered by demographics (Table 2) and appropriate except IQR is not explained in the footnotes to the table. What is IQR?
i. Anticholinergic drug use at admission and discharge is described but a table would also be helpful to show which drugs (though not required).
ii. The Figure Title for figure 3 printed on the next page of the pdf I used so hopefully the editor will catch this and get them on the same page. Unfortunately the figure appears to come from a program and could be difficult to reduce in size.
iii. Curiously the authors discuss the 30-day readmission but do not discuss the length of stay. Please add before the readmission data.
5. Discussion: Logical and easy to follow except for the length of stay outcome. I would encourage the authors to also consider that the use of the term “dementia” which is non-disease specific and may be why so many were treated with cholinesterase inhibitors which is really disease specific treatment (i.e. Alzheimer’s Disease) but clinically is often prescribed by primary care providers who may not have actually determined what type of dementia the person has Mild cognitive impairment cannot be treated by any known drug, nor can any form of dementia be “prevented” with a drug to date.
a. Please add why you think the LOS did not change.
6. Limitations are quite good and address the issue with medications for COPD.
7. References are appropriate.
8. Minor edits
a. Line 158 change word “clinical” to “clinically” relevant.
b. Table 1 Consider adding Column headings if this table covers more than one page so the reader doesn’t have to flip back a page.
Comments on the Quality of English Language
Minor. In the comments to the authors.
Author Response
Brief Summary: This is a process improvement project in which an updated scale was developed to estimate the anticholinergic burden of hospitalized geriatric patients in Denmark. Secondary aims were to examine if any change occurred in this burden at discharge, and the relationship to length of stay and 30-day hospital readmission.
General Comments: This manuscript flowed logically and is generally well written with one major concern and a few minor comments. The major concern is the lack of comment about the length of stay and anticholinergic burden which is listed a secondary outcome measure.
Specific Comments
Comment 1: Abstract is clear but should also include the length of stay finding.
Response 1: A sentence about lenght of stay have been included in the abstract, line 22.
Comment 2: Introduction is clear and logical concluding with the objectives of the study: burden at admission and discharge (and any change), length of stay, and 30-day readmission.
Response 2: We appreciate your comment.
Comment 3: Materials and Methods:
Development of the Updated Scale: This process is adequately described and appropriate literature used to create the list of 87 drugs. This reviewer was particularly impressed with the inclusion of receptor binding affinities criterion and other descriptions of how the 87 drugs were selected. This alone makes the scale clinically relevant to the institution where used. The simplicity of the scale also is appealing as a clinically relevant tool.
The Clinical Assessment: Appropriate and clear participant selection and exclusion as well as the primary and secondary outcomes to be measured.
Data Analysis: Appropriate for level of data.
Response 3: We appreciate your comment.
Comment 4: Results section with tables
The updated scale: Table 1 is clear and concise as are the comments about it.
Comment 4a: The Clinical Assessment results are clustered by demographics (Table 2) and appropriate except IQR is not explained in the footnotes to the table. What is IQR?
Response 4a: Thank you for pointing
Comment 4b: Anticholinergic drug use at admission and discharge is described but a table would also be helpful to show which drugs (though not required).
Response 4b: Thank you for the suggestion, as this is not required we have diced not to include it.
Comment 4c: The Figure Title for figure 3 printed on the next page of the pdf I used so hopefully the editor will catch this and get them on the same page. Unfortunately the figure appears to come from a program and could be difficult to reduce in size.
Response 4c: A note for editor.
Comment 4d: Curiously the authors discuss the 30-day readmission but do not discuss the length of stay. Please add before the readmission data.
Response 4d: We have added a point regarding data on length of stay.
Comment 5: Discussion: Logical and easy to follow except for the length of stay outcome. I would encourage the authors to also consider that the use of the term “dementia” which is non-disease specific and may be why so many were treated with cholinesterase inhibitors which is really disease specific treatment (i.e. Alzheimer’s Disease) but clinically is often prescribed by primary care providers who may not have actually determined what type of dementia the person has Mild cognitive impairment cannot be treated by any known drug, nor can any form of dementia be “prevented” with a drug to date.
Response 5: We used the term dementia due to the diagnosis documented in the electronic chart but thanks for emphasizing the issue.
Comment 5a: Please add why you think the LOS did not change.
Response 5a: Length of stay maybe influenced by issues not related to anticholinergic burden. E.g. patients’ support at home have to be arranged before discharging the patients including ordering of care, meals, etc.
Comment 6: Limitations are quite good and address the issue with medications for COPD.
Response 6: Thank you for your comment, it is appreciated.
Comment 7: References are appropriate.
Response 7: The comment is appreciated.
Comment 8: Minor edits
- Line 158 change word “clinical” to “clinically” relevant.
- Table 1 Consider adding Column headings if this table covers more than one page so the reader doesn’t have to flip back a page."
Response 8: Thank you for the suggestions. We have addressed the first, while we leave the second to be addressed by the editor.